# Ramosin: The First Antibacterial Peptide Identified on *Bolitoglossa ramosi* Colombian Salamander

**DOI:** 10.3390/pharmaceutics14122579

**Published:** 2022-11-23

**Authors:** Laura Medina, Fanny Guzmán, Claudio Álvarez, Jean Paul Delgado, Belfran Carbonell-M

**Affiliations:** 1Grupo Genética, Regeneración y Cáncer, Facultad de Ciencias Exactas y Naturales, Instituto de Biología, Universidad de Antioquia, Medellín 050010, Colombia; 2Núcleo de Biotecnología Curauma (NBC), Pontificia Universidad Católica de Valparaíso, Valparaíso 2373223, Chile; 3Laboratorio de Fisiología y Genética Marina (FIGEMA), Centro de Estudios Avanzados en Zonas Áridas (CEAZA), Coquimbo 1781421, Chile; 4Facultad de Ciencias del Mar, Universidad Católica del Norte, Coquimbo 1781421, Chile; 5Departamento de Estudios Básicos Integrados, Facultad de Odontología, Universidad de Antioquia, Medellín 050010, Colombia

**Keywords:** antibacterial, peptide, salamander, bioinformatics

## Abstract

The discovery and improvements of antimicrobial peptides (AMPs) have become an alternative to conventional antibiotics. They are usually small and heat-stable peptides, exhibiting inhibitory activity against Gram-negative and Gram-positive bacteria. In this way, studies on broad-spectrum AMPs found in amphibians with the remarkable capability to regenerate a wide array of tissues are of particular interest in the search for new strategies to treat multidrug-resistant bacterial strains. In this work, the use of bioinformatic approaches such as sequence alignment with Fasta36 and prediction of antimicrobial activity allowed the identification of the Ramosin peptide from the de novo assembled transcriptome of the plethodontid salamander Bolitoglossa ramosi obtained from post-amputation of the upper limb tissue, heart, and intestine samples. BLAST analysis revealed that the Ramosin peptide sequence is unique in Bolitoglossa ramosi. The peptide was chemically synthesized, and physicochemical properties were characterized. Furthermore, the in vitro antimicrobial activity against relevant Gram-positive and Gram-negative human pathogenic bacteria was demonstrated. Finally, no effect against eukaryotic cells or human red blood cells was evidenced. This is the first antibacterial peptide identified from a Colombian endemic salamander with interesting antimicrobial properties and no hemolytic activity.

## 1. Introduction

Antimicrobial peptides (AMPs) are biologically active molecules produced by a wide variety of organisms, such as bacteria, yeast, fungi, plants, and animals, as an essential component of their innate immune response [1,2,3]. AMPs play a critical role as defense molecules to protect animals and plants from the invasion of viruses, fungi, or bacterial pathogens [4,5]. These peptides have the ability to eliminate the invading pathogenic microorganisms directly, and some of them also serve as immune modulators in higher vertebrates [6]. In addition, due to their broad range of activity, less toxicity, specificity, and decreased resistance development by the target cells, AMPs are considered for the near future as promising antimicrobial drug candidates [5,7,8].

These peptides can be constitutively or inducibly expressed depending on the organism and the tissue in which they are present at the time of infection [9]. Exocrine glands present in the skin of many species of anurans (frogs, toads) and salamanders secrete huge amounts of a rich variety of hormones and neuropeptides belonging to families of peptides that have their counterparts in human brain and gut [10]. Exocrine glands can release their contents to the skin surface by a holocrine mechanism involving rupture of the plasma membrane and extrusion of secretory granules through a duct that opens to the surface [11]. The serous glands of anurans and salamanders also produce a rich arsenal of AMPs that are thought to be involved in defending bare skin against harmful microorganisms and to aid in wound repair [12].

Amphibians exhibit a complex system of AMPs in their immune system to avoid pathogenic microorganisms that inhabit the damp and dark environment where they thrive [13,14,15]. Amphibians are widely distributed across numerous habitats worldwide and comprise the oldest class of animals from which AMPs compounds have been isolated [16,17]. Most amphibians produce secretions that are released onto their skin and have been widely identified from anurans’ skins but not in salamanders [17,18,19].

Salamanders belong to the order of Urodele, which corresponds to 9% of amphibians. Adult salamander skin consists of two layers of tissue: a stratified keratinized epidermis and an underlying thick layer of collagen-rich dermal connective tissue that provides mechanical support and nutrition [19]. They present a wide variety of antipredator mechanisms, including toxins and noxious or adhesive skin secretions [20]. Their skin glands secrete poison and antibiotics to protect themselves from predators and against infection by microbial pathogens; salamander skins are a chemical matrix that acts as a defensive mechanism [21]. *Bolitoglossa ramosi* (*B. ramosi*) is a lungless endemic salamander species of Colombia, belonging to the Plethodontidae family. It is a nocturnal salamander that inhabits Andean and sub-Andean forests, with an altitude distribution between 1200 and 2000 m above sea level, and this distribution is restricted to the eastern flank of the central mountain range [22,23,24].

In a previous work made by Arenas [23], a total of nine adults of *B. ramosi* underwent limb amputations, and tissues from the limb, gut, and skin were collected at the time of amputation and at 20, 40, 60, and 70 days post-amputation. The reference transcriptome was assembled from all RNA samples, and contigs with a length of ≥200 nucleotides were extracted to generate an initial reference transcriptome. Arenas identified 109 genes that were differentially expressed throughout regeneration in *B. ramosi,* of which 77 were previously identified as differentially expressed during limb regeneration in *Ambystoma mexicanum* [25]. Repair of tissue wounds is a fundamental process to re-establish tissue integrity and regular function. Importantly, infection is a major factor that hinders wound healing. Multicellular organisms have evolved to produce an arsenal of host-defense molecules, including AMPs, aimed at controlling microbial proliferation and at modulating the host’s immune response to a variety of biological or physical injuries [26,27,28,29].

The discovery and design of AMPs have been boosted during the post-genomic era. Pattern matching and sequence alignment with peptides from different databases have been intensively employed for the identification of unannotated AMPs. Recently, the explosive growth of data sequencing has stimulated the application of powerful machine learning algorithms in biomedical areas, including genomic mining and design of AMPs [30,31]. However, the physicochemical characteristics of the AMPs are not completely elucidated, resulting in issues regarding how new AMPs can be identified from transcriptomes, in this way reducing cost and time, avoiding the need to kill the animals [32]. Currently, some databases compile information on these peptides, such as the Antimicrobial Peptide Database 3 (APD3) [33], which contains 3425 AMPs from six life kingdoms (385 isolated/predicted bacteriocins/peptide antibiotics from bacteria, 5 from archaea, 8 from protists, 25 from fungi, 368 from plants, 2489 from animals, and some synthetic peptides), including 1148 active peptides identified on amphibians (1070 from frogs and 74 from toads), with only three of the amphibian peptides reported from salamanders. Those three peptides were identified in the Chinese giant salamander *Andrias davidianus* [34]. This work was focused on identifying AMPs from transcriptome sequences obtained from post-amputation tissues, heart, and intestine from *B. ramosi* using bioinformatics analysis. Ramosin peptide was identified by alignment with the peptides from APD3, and then was chemically synthesized. In vitro tests against Gram-positive and Gram-negative bacteria demonstrated the antimicrobial properties of this peptide with lack of hemolytic activity. In addition, the secondary structure and possible action mechanism of Ramosin on bacterial membranes were analyzed. This peptide is the first identified in the Plethodontidae amphibia family.

## 2. Materials and Methods

### 2.1. Bioinformatic Analysis

#### 2.1.1. Transcriptome

The bioinformatic analyses were carried out from the reference transcriptome of *Bolitoglossa ramosi* that was assembled de novo in 2018 by Arenas et al. (NCBI with the access code GSE105232) [25]. This transcriptome was assembled using sequences obtained from a group of nine adult animals and consists of sequences from intestine, heart, skin, and blastema samples at 20, 40, 60, and 70 days post-amputation of the upper limb. The de novo transcriptome of *B. ramosi* was performed using end-matched Illumina sequencing technology and assembled with Trinity [35]. A total of 433,809 transcripts were recovered and functional annotation was made for 142,926 nonredundant transcripts [25]. The identification of the potential AMPs was carried out by an alignment between peptide sequences from cured databases and the de novo transcriptome of *B. ramosi* made with the pool of samples from regenerative tissues, heart, and intestine.

#### 2.1.2. Protein Prediction from the De Novo Transcriptome of *Bolitoglossa ramosi*

The TranDecoder-v5.4.0 tool was used to carry out the translation to putative proteins from the transcriptome that was assembled into nucleotides. The hypothetical proteins that are encoded in the *B. ramosi* transcriptome were predicted from the six possible and longest Open Reading frames (available online: https://github.com/TransDecoder/TransDecoder/wiki (accessed on 28 December 2020)).

#### 2.1.3. APD3 Peptide Database

APD3 (available online: https://aps.unmc.edu/) is one of the largest cured peptide databases (last accessed on 10 October 2022) [34]. This original database for AMPs is manually curated based on a set of data-collection criteria. There are 3425 AMPs from six life kingdoms; 1148 of those active peptides are from amphibians, 1070 from frogs, 74 from toads, and only three of them are from salamanders: the AP02897 (AdCath), the AP02922 (Andricin 01), and the AP02923 (Andricin B). APD3 provides searchable annotations including source organism, peptide sequence, principal physicochemical properties, and structural classification [34]. The sequences of AMPs from APD3 were used for subsequent alignments with the hypothetical proteins from the *B. ramosi* transcriptome.

#### 2.1.4. Alignments to Identify Candidate Peptides

To identify potential AMPs on *B. ramosi,* sequences from the APD3 database were aligned with hypothetical proteins obtained from the transcriptome of *B. ramosi,* using a global–local alignment with Fasta36 software [36,37]. The algorithm used for the protein–protein alignment was glsearch36 with an E-value threshold of 1 × 10^−3^, and the one showing the five better alignments with the best score was used to select the best match (available online: https://github.com/wrpearson/fasta36) (accessed on 28 December 2020). With the purpose of finding sequences not previously reported in the literature, the sequences that had a similarity of >80% were selected [32,38]. Those parameters were chosen in order to have sequences with a high probability of having the same secondary structure and antimicrobial activity of the original peptide sequences from the APD3 database.

#### 2.1.5. In Silico Prediction of the Antimicrobial Activity of Candidate Peptides

To determine if candidate peptides of *B. ramosi* had potential antimicrobial activity, the ClassAMP predictor that works by the model of support vector machine was used (available online: http://www.bicnirrh.res.in/classamp/predict.php) (accessed on 28 December 2020) [39]. This tool provides a probability value between 0 and 1, indicating the possible activity, or lack of it, of the peptides that were selected as candidates.

#### 2.1.6. Molecular Confirmation of the Candidate Peptides

To verify the presence of the coding sequences for the candidate peptides, the cDNA of *B. ramosi* was amplified by PCR, using specific primers for the region of the mature peptide, and the PCR products were sequenced by the Sanger method. The cDNA samples to verify the presence of the sequences encoding the peptides were blastemas, skin, and intestine tissues. Those samples were collected by Arenas in a previous work [23]. The list of the primers and PCR conditions are in the Appendix A.

### 2.2. Peptide Synthesis and Characterization

Candidate peptides were synthesized by solid-phase synthesis by using the standard Fmoc/tBu strategy, using tea bags according the protocol developed by Houghten [40]. Amino acids and activators were acquired from Iris Biotech GmbH (Marktredwitz, Germany), and solvents, deprotection, and cleavage reagents were acquired from Merck KGaA (Darmstadt, Germany). Synthesis was performed according to previous work [41,42]. Briefly: tea bags with 40 mg of Fmoc-Rink Amide AM resin with a substitution of 0.5 meq/g were used. Coupling cycles were performed by deprotecting the Fmoc group with 4-methyl piperidine at 20% in N,N-dimethylformamide (DMF), and then by adding the corresponding amino acid solution, using N-[(1H-benzotriazol-1-yl)-(dimethylamino)methylene]-N-methylmethanaminium hexafluorophosphate N-oxide (HBTU) and N-[6-chloro(1H-benzotriazol-1-yl)-(dimethylamino)methylene]-N-methylmethanaminium hexafluorophosphate N-oxide (HCTU) as activators for the first and second coupling, respectively; additionally, N-ethyldiisopropylamine (DIPEA) was used for neutralization in the coupling solution. To verify the end of each coupling, a solution of bromophenol blue at 0.1% in DMF was used. After each step, washes were performed with DMF and the final cleavage of the peptide was carried out with a solution of trifluoroacetic acid (TFA), triisopropylsilane (TIS), and water in proportion 95:2.5:2.5 and with TFA:TIS:water:2.2′-(ethylenedioxy) diethanethiol (DOT) (92.5:2.5:2.5:2.5); in the case of peptides that contained Cys, Met, or Trp residues. After cleavage, peptides were precipitated with cold diethyl ether, centrifugated, and washed five times with diethyl ether. Peptides were dried and dissolved in Milli-Q water, then frozen and lyophilized.

#### 2.2.1. Determination of the Main Fraction Containing the Expected Peptide

The peptide’s purity was verified by reverse-phase high-performance liquid chromatography (RP-HPLC), which separates the molecules based on their hydrophobicity. A Jasco chromatograph with UV-2075 Plus detector, PU-2089 Plus quaternary pump, and AS-2055 Plus autosampler was used, with a Water Corp XBridge BEH130 C18 3.5 μm dp, 4.6 × 100 mm column. The peptides were analyzed using a gradient of 0–70% acetonitrile (ACN) for 8 min and detection at 220 nm. The retention times of the crude peptides were determined, and purification was subsequently carried out on Clean-Up^®^CEC18153 C-18 columns (UCT, Bristol, PA, USA). The peptides were eluted according to the percentages of ACN observed in the analysis with the RP-HPLC. Next, the ACN solvent was removed from samples with a SpeedVac^®^ Concentrator (Thermo Fisher, Waltham, MA, USA), and then the peptides were frozen and lyophilized.

The characterization of the molecular weight of the purified peptides was carried out in a UFLC-ESI Shimadzu LCMS-2020 equipment, using a 0–100% ACN gradient in 20 min. The peptide was injected and dissolved in water.

#### 2.2.2. Circular Dichroism

The analysis of the conformation of the Ramosin peptide was performed through circular dichroism (CD). CD spectroscopy was carried out on a JASCO J-815 CD Spectrometer (JASCO Corp., Tokyo, Japan) in the far ultraviolet (UV) range (190–250 nm), using quartz cuvettes (0.1 cm path length). Each CD spectra of the synthetic peptide was recorded, averaging three scans in continuous scanning mode. Solvent blank was subtracted from each sample spectrum. Molar ellipticity was calculated for each spectra using 250 µL of 2 mM peptide in 30% (*v*/*v*) 2,2,2-trifluoroethanol (TFE), water, phosphate buffered saline (PBS) 2 mM pH = 7.4, sodium bicarbonate buffer 1 M pH = 8.5, and sodium acetate 3 M pH = 5. Resulting data were analyzed using Spectra Manager software (Version 2.0, JASCO Corp., Tokyo, Japan) [41,42].

### 2.3. Antimicrobial Activity

To determine the minimum inhibitory concentrations (MICs) of the candidate peptides, the 96-well plate microdilution assay, described by the Clinical and Laboratory Standards Institute (CLSI) and the European Committee on Antimicrobial Susceptibility Testing (EUCAST) [43], was performed. For testing the antimicrobial activity of the candidate peptides, Gram-positive and Gram-negative bacterial strains were used. The strains used were *Staphylococcus aureus* ATCC 29,213 *(S. aureus)*, *Enterococcus faecalis* ATCC 29,212 (*E. faecalis)*, *Bacillus cereus* ATCC 6464 *(B. cereus),* and *Micrococcus luteus* ATCC 9341 *(M. luteus)* as Gram-positive, and *Escherichia coli* ATCC 25,922 *(E. coli), Pseudomonas aeruginosa* ATCC 27,853 (*P. aeruginosa*), and *Salmonella typhimurium* ATCC 14,028 (*S. typhimurium*) as Gram-negative. The bacteria were cultivated in tryptic soy agar (TSA) until the time of carrying out the antimicrobial activity assay. For this assay, 3 to 5 colonies were taken from the agar and grown to exponential phase in tryptic soy broth (TSB) liquid medium defined as a 0.5 McFarland value measured with a densichek plus (Biomérieux). Bacteria were diluted to a final concentration of 5 × 10^5^ CFU/mL. *E. coli*, *P. aeruginosa*, *E. faecalis*, and *B. cereus* were treated with Ramosin peptide at 4.4, 8.7, 17.5, 35, and 70 μM, and *S. aureus* were treated with 5, 10, 20, 40, and 50 μM. The plates were incubated for 18 h at 37 °C, and subsequently the absorbance was read in a MultiSkan GO (Thermofisher, Waltham, MA, USA) at 600 nm. Each well was tested in triplicate and each experiment was performed in three independent runs. To determine if the effect of the peptides was bactericidal or bacteriostatic, once the absorbance reading was completed, 10 μL were taken from each well where bacterial growth inhibition was observed and they were sown in TSA agars and incubated at 37 °C for 24 h. Once this time had elapsed, it was verified whether or not bacterial growth had occurred [44].

### 2.4. Hemolytic Assay

The hemolytic activity of the peptide was determined as previously reported [45]. In brief, its hemolytic activity of the peptide was evaluated by determining the amount of the released hemoglobin from a 4% *v/v* suspension of fresh human red blood cells (RBCs). In 96-well plates, 65 μL of the RBCs suspension and 65 μL of each peptide dilution were mixed. RBCs were treated with Ramosin peptide at 4, 8, 16, 32, 64, and 128 μM and with PBS1x and Triton X-100 at 0.5% *v/v* as negative and positive controls of hemolysis. Plates were incubated at 37 °C for two hours, then centrifuged at 2500× *g* for 5 min. A total of 80 μL of supernatant was recovered and transferred to another 96-well plate, and absorbance at 540 nm was measured in a MultiSkan GO (Thermofisher, Waltham, MA, USA). The percentage of hemolysis was calculated according to the following formula [45]:
(1)$$\%\mathrm{Hemolysis}=\frac{\mathrm{OD}450\;\mathrm{in}\;\mathrm{the}\;\mathrm{peptide}\;\mathrm{suspension}-\mathrm{OD}450\;\mathrm{negative}\;\mathrm{control}\;\mathrm{in}\;\mathrm{PBS}}{\mathrm{OD}450\;\mathrm{of}\;\mathrm{positive}\;\mathrm{control}-\mathrm{OD}450\;\mathrm{negative}\;\mathrm{control}\;\mathrm{in}\;\mathrm{PBS}}\times 100$$

### 2.5. Scanning Electron Microscopy (SEM)

Aliquots of *E. coli* ATCC 25,922 and *S. aureus* ATCC 29,213 were harvested in logarithmic growth phase, and then centrifuged at 1000× *g* for 5 min. Cell pellets were washed twice with 10 mM PBS and resuspended in the same buffer. The bacteria were incubated at 37 °C for 20 min with the peptides at different concentrations (20 or 30 μM). Peptides were diluted in sterile Milli-Q water. After incubation, the bacteria were centrifuged and washed three times with PBS, each time centrifuging at 1000× *g* for 5 min. After washing, the bacterial cells were placed on glass coverslips and fixation was carried out with 2.5% (*v*/*v*) glutaraldehyde in PBS for 20 min at room temperature. Once the bacterial cells were fixed, a gradual dehydration with ethanol was carried out, critical point drying being carried out using the Samdri-780A equipment (Tousimis Research Corporation, Rockville, MD, USA), and the slides were vacuum coated using a platinum and palladium plate to avoid charging in the microscope (fine coat, ion sputter JFC-1100). To visualize the bacteria, a Hitachi SU 3500 scanning electron microscope was used [46].

### 2.6. Cell Lines

Three cell lines were used: one nontumor cell line, HaCat, a spontaneously transformed human keratinocyte, and two cell lines of tumorigenic origin. MCF-7 (ATCC: HTB22) is a cell line derived from a metastatic site of human mammary adenocarcinoma and PC-3 (ATCC: CRL-1435) is derived from a grade IV adenocarcinoma of the human prostate. T25 cell culture flasks (Falcon, reference: 353018) with Dulbecco’s Modified Eagle’s Medium (DMEM) (Gibco, reference: 12100046) supplemented with fetal bovine serum at 5% (FBS, Gibco, reference: 12657029), penicillin 100 U/mL, and streptomycin 100 μg/mL (Gibco, reference: 15140122) were used for the maintenance of cell cultures in monolayer. Cells were incubated at 37 °C in the presence of 5% CO_2_.

### 2.7. In Vitro Cytotoxicity Assay

Cells were cultured in 96-well plates (7 × 10^3^ cells/well) in DMEM with 5% FBS. After 24 h of incubation, cells were treated with Ramosin peptide (0.5, 5, and 50 μM) and incubated for another 24 h. Cell viability was measured with the 3-(4,5- dimethylthiazol-2-yl)-2,5-diphenyl tetrazolium bromide (MTT, Sigma, reference: M5655) assay. Cells that were not treated were used as negative control and wells containing only medium and 5% FBS were left as blanks of the experiment. As a positive control, the cells were treated with hydrogen peroxide (H_2_O_2_) at 30 μM. The treatments with peptides at 0.5, 5, and 50 μM were carried out for 24 h at 37 °C and 5% CO_2_. After this time, 10 μL of MTT (5 mg/mL) per well were added, and the plate was incubated at 37 °C and 5% CO_2_ for 3 h. After this, 100 μL of isopropanol acid were added to each well, and the formazan crystals were dissolved with gentle stirring. The absorbance reading in the plate was performed in a MultiSkan GO at 570 nm. Finally, the cell viability percentage of each well was obtained, comparing treated cells and controls without treatment.

### 2.8. Statistic Test

The comparison of the percentages of viability obtained after the treatment of the bacteria and the mammalian cells with the different concentrations of the evaluated peptides versus the controls, were carried out through the Kruskal–Wallis H-tests and multiple comparisons through the Mann–Whitney U-test, with Bonferroni correction, or one-factor ANOVA and multiple comparisons with Tukey’s HSD test. In all cases, compliance with the assumption of normal distribution and homogeneity of variances was previously verified with the Shapiro–Wilk and Levene statistics, respectively. In all the analyses, a *p* value of statistical significance less than 0.05 was taken as the criterion for accepting or rejecting the null hypothesis. The analyses were performed using the programs IBM^®^ SPSS 27 (IBM Corp. Released 2020. IBM SPSS Statistics for Windows, Version 27.0. Armonk, NY, USA: IBM Corp) and GraphPad Prism version 6.04 (GraphPad Software, La Jolla, CA, USA).

## 3. Results

### 3.1. In Silico Identification of AMPs from B. ramosi Salamander

A total of 82,122 hypothetical proteins from *B. ramosi* transcriptome were obtained with TransDecoder. Global–local alignments with Fasta36 were chosen to avoid results with partial alignments. Once the alignment was completed with the peptides from the APD3 database, 23 peptides with 100% identity were found, corresponding to peptides already reported. These peptides were between 6 and 130 amino acids and were identical to the sequences of the mature peptides. Table 1 shows the peptides from APD3 that present 100% identity with hypothetical proteins of *B. ramosi.* These peptides were both cationic and anionic, and exhibited different functions. The most common function was antibacterial, but some of them also had antifungal, antiparasitic, antiviral, and anticancer activity. This is the first report of possible multifunctional peptides on the *B. ramosi* salamander. Future investigations are needed to probe the real expression of those peptides during the wound-healing process.

To search for potential new AMPs from the *B. ramosi* transcriptome, peptide sequences with a similarity percentage, calculated with Fasta36, greater than 80% were selected and considered as candidates. A total of 91 sequences with ≥80% similarity were identified, and 59 sequences were unique. Of these, only 24 presented potential antimicrobial activity according to the ClassAMP predictor [39]. For the peptide synthesis, 11 sequences with a length ≤ 21 amino acids were selected. Once the 11 candidate peptides had been identified, the sequences encoding these peptides were molecularly verified by conventional PCR and subsequent sequencing in Macrogen, in order to demonstrate that the sequences were not artifacts of the bioinformatics tools used (Appendix B). The 11 sequences of candidate peptides that were synthesized are shown in Table 2.

Once the chemical synthesis of the candidate peptides was finished, cleavage and lyophilization were carried out. Later on, purity analyses were carried out by RP-HPLC and the molecular weight was corroborated by electrospray ionization mass spectrometry (ESI MS) (Appendix A). Peptide 3428 did not have the expected mass; therefore, it was excluded from subsequent experiments. The other candidate peptides were used for the in vitro antimicrobial activity test.

The first antimicrobial activity that was carried out was a screening with Gram-negative bacteria (*E. coli and S. typhimurium*) and Gram-positive bacteria (*S. aureus, M. luteus, and B. cereus*) to determine which of the synthesized peptides had antimicrobial activity (Appendix B) (Appendix A). Comparisons of the percentages of viability of the microorganisms evaluated at the different concentrations of the candidate peptides and the control without treatment were made by means of a one-factor ANOVA and multiple comparisons with the Tukey’s HSD test, and it was found that peptide 3412 decreases the viability percentage of *E. coli* (*p* < 0.05, 95% CI 23.4132 to 46.9912), *S. typhimurium* (*p* < 0.05, 95% CI 30.2223 to 57.7228), *S. aureus* (*p* < 0.05, 95% CI 29.4628 to 54.8291), *M. luteus* (*p* < 0.05, 95% CI 67.0026 to 94.6356), and *B. cereus* (*p* < 0.05, 95% CI 35.5268 to 60.7372). The other candidate peptides evaluated did not show a statistically significant decrease in the percentage of viability of the Gram-negative and Gram-positive bacteria evaluated. For this reason, we continue the additional tests of in vitro activity only with peptide 3412.

As candidate peptide 3412 was identified from the transcriptome of the endemic Colombian salamander *B. ramosi,* we decided to call it Ramosin. From now on, peptide 3412 will be referred as Ramosin peptide.

### 3.2. Assays of Antibacterial Activity of Ramosin Peptide from B. ramosi

#### 3.2.1. Antibacterial Assay of Ramosin Peptide

All bacterial strains used were ATCC reference strains. The antibacterial activity of Ramosin was evaluated using the Wiegand et al. protocol [43]. Buforin II was used as control peptide. Two types of tests were carried out: the endpoint, in which the absorbance of the plates was read after 19 h of treatment, and kinetics, in which plate readings were made every hour to determine whether the effect of the peptides was bactericidal or bacteriostatic.

In the endpoint tests, it was observed that Ramosin had an antibacterial effect on both strains of Gram-negative bacteria. In these endpoint assays, comparisons were made of the percentages of viability of the bacteria evaluated and the different concentrations of Ramosin, using Kruskal–Wallis H-tests and multiple comparisons using Mann–Whitney U-tests with Bonferroni correction. For *E. coli,* it was observed that there is a decrease in the percentage of viability of the bacteria after treatment with the peptide from a concentration of 17.5 μM. This decrease is statistically significant for the concentrations of 17.5 μM, 35 μM, and 70 μM (*p* < 0.05). The inhibition observed is comparable to that obtained with the treatment with NaClO 0.012% (*p* < 0.05) or with Buforin II at 70 μM (*p* < 0.05) (Figure 1a).

For *P. aeruginosa,* it was observed that the inhibitory effect as a consequence of the treatment with the Ramosin peptide occurs from 35 μM to 70 μM, again matching the results obtained with NaClO 0.012% (*p* < 0.05) or with Buforin II at 70 μM (*p* < 0.05) (Figure 1b). This decrease is statistically significant for the concentrations of 35 μM (*p* < 0.05) and 70 μM (*p* < 0.05).

In addition to absorbance measurement, a bacterial growth test was performed. Once the 19 h treatment was over, 10 μL were taken from the wells where growth inhibition was seen and they were sown on TSA to see if bacterial growth occurred after 24 h of incubation at 37 °C. In cases where no growth was observed after 24 h of incubation, that peptide concentration was determined to be bactericidal, and if growth was observed it was determined as a bacteriostatic concentration.

In the kinetic experiments, it was observed that the Ramosin peptide had a bactericidal effect against *E. coli* from a concentration of 17.5 μM on (*p* < 0.005), and its effect was maintained for at least the 19 h of the test (Figure 2a). Regarding *P. aeruginosa*, peptide Ramosin presented a bactericidal effect from the concentration of 35 μM on (*p* < 0.005), and a bacteriostatic effect in the treatment at the concentration of 17.5 μM (*p* < 0.005), since an inhibition in the growth of bacteria was observed during the first 11 h of the test; but once this time had elapsed, the bacteria began to grow again (Figure 2b). This explains why, in the endpoint assay, an inhibitory effect on the growth of *P. aeruginosa* treated with the Ramosin peptide at a concentration of 17.5 μM was not observed (Figure 2b). It can also be seen that Buforin II had a bactericidal effect at a concentration of 70 μM (*p* < 0.005) on both Gram-negative bacteria, and that this effect was observed from the beginning of the test until the 19th h of treatment (Figure 2).

In the endpoint assay, for *S. aureus* there was growth inhibition only in the treatment with Ramosin peptide at concentrations of 40 μM and 50 μM. This decrease is statistically significant only for the 50 μM concentration (*p* < 0.005). The inhibition observed is comparable to that obtained with the treatment with NaClO 0.012% (*p* < 0.005). The Buforin II peptide, which was being used as a peptide control, worked very well against Gram-negative bacteria; however, when its effectiveness against Gram-positive bacteria was tested, it did not have a statistically significant effect (*p* > 0.005) (Figure 3a).

For *E. faecalis,* the only treatment that caused a statistically significant decrease in the percentage of viability was the positive control for death, NaClO 0.012% (*p* < 0.005). Therefore, the MIC for Ramosin in *E. faecalis* should be greater than 70 μM (*p* > 0.005) (Figure 3b). For *B. cereus,* it was observed that the treatment with the peptide Ramosin at a concentration of 70 μM (*p* < 0.005) produced an inhibition of bacterial growth similar to that caused by the treatment with NaClO at 0.012% (*p* < 0.005) (Figure 3c).

For the kinetics of bacterial growth in the strains of Gram-positive bacteria treated with the Ramosin peptide, it was observed that the treatment of *S. aureus* with Ramosin at 50 μM and 40 μM presented a bactericidal effect during the 19 h of the test; while, at the concentration of 20 μM, a bacteriostatic effect was obtained during the first 10 h of treatment (*p* < 0.005) (Figure 4a). In *B. cereus,* it was observed that the concentration of 70 μM of the Ramosin peptide was bactericidal and that the concentration of 35 μM of said peptide presented a bacteriostatic effect during the first 9 h of the test (*p* < 0.005) (Figure 4b). Similarly, the concentration of 17.5 μM (*p* < 0.005) inhibited growth during the first 7 h of the kinetic assay; this may explain why there was an inhibitory effect on the growth of *B. cereus* treated by 19 h with these concentrations of Ramosin peptide (Figure 4b). In both tests, it was observed that Buforin II at a concentration of 70 μM failed to inhibit the growth of Gram-positive bacteria at any time during the 19 h evaluated.

#### 3.2.2. Hemolytic Assay of Ramosin Peptide

Once it was determined that the Ramosin peptide had antimicrobial activity inhibiting the growth of both Gram-positive and Gram-negative bacteria, we proceeded to evaluate whether it had any effect on human red blood cells. The hemolysis assay shows that the Ramosin peptide is not hemolytic (Appendix A), at least up to a concentration of 128 μM, which is higher than the maximum antimicrobial concentration obtained for the Ramosin peptide to inhibit Gram-negative and Gram-positive bacteria.

#### 3.2.3. Scanning Electron Microscopy Assay with Ramosin Peptide

In order to assess whether the antimicrobial effect of the Ramosin peptide was due to interaction with the membranes of Gram-positive and Gram-negative bacteria, we performed a scanning electron microscopy analysis with *E. coli* and *S. aureus*. Figure 5 shows that the Ramosin peptide did not cause a significant disturbance on the surface of *E. coli*, when comparing bacteria treated with this peptide and untreated bacteria. On the contrary, the treatment of *E. coli* with the control peptide BTM-P1 [47] causes its complete lysis, showing that it has a powerful lysogenic effect.

In Figure 6 it can be seen that the Ramosin peptide caused a disturbance in the membranes of *S. aureus* since protuberances were observed, suggesting that the possible mechanism of action of the Ramosin peptide in *S. aureus* is an interaction with its membrane.

#### 3.2.4. In Vitro Cytotoxicity Assay with MTT

MTT was used as a screening test with which the cytotoxicity of peptides can be determined. The redox potential in viable mammalian cells causes the water-soluble MTT reagent to convert to an insoluble formazan product. A total of 30 μM hydrogen peroxide was used as a positive control for cytotoxicity. Figure 7 shows that the Ramosin peptide did not cause a statistically significant reduction in the percentage of viability of the three cell lines evaluated, HaCat, MCF-7, and PC-3, after being treated for 24 h with peptide concentrations of 0.5 μM, 5 μM, and 50 μM. Subsequently, flow cytometry analyses were performed to measure mitochondrial membrane potential involvement with 3,3’-Dihexyloxacarbocyanine iodide (DiOC_6_) and propidium iodide (PI) and to assess cell membrane integrity and apoptosis with Annexin V and Sytox, with assays of cells treated for up to 48 h with concentrations up to 200 μM of Ramosin. However, no cytotoxic effect was observed (Appendix A).

#### 3.2.5. Molecular Characterization of Ramosin Peptide

When Ramosin was identified as an antibacterial nonhemolytic peptide, a molecular characterization of the sequence was carried out. The distribution of the reads in the sequence was evaluated and the number of reads was determined according to the tissue of origin [25]. The number of reads vary between the specimens: some of them showed more expression of Ramosin peptide in the limb and others showed more expression in the intestine. An interesting fact is that Ramosin could be expressed in different tissues. Table 3 shows the number of reads per type of sample and the percentage of discordance between reads.

The number of reads supported by the transcripts encoding the Ramosin peptide was validated. Since the *B. ramosi* transcriptome was assembled from samples of different tissues, we verified the presence of these transcripts in skin, intestine, limb, and blastema. We found that the transcripts were well supported by the number and distribution of reads, showing that they are not the product of a bioinformatics artifact. Appendix A shows the distribution of the reads on the sequence of nucleotides that codify for the Ramosin peptide. This analysis was performed because the sequence of the Ramosin peptide is unique (it did not have a match after Blast analysis).

The Ramosin peptide was similar to the Aureins peptides reported on the APD3 database [48]. Figure 8 shows the alignment of the Ramosin peptide and the sequences of the Aureins reported in APD3 with which it presented a similarity of ≥80%. Aurein sequences were identified in 2000 by Rozek et al. and correspond to peptide sequences with antibiotic and antitumor functions, which are produced in the skin of the frog *Litoria aurea* [49]. In Figure 8 it is possible to identify some conserved amino acids between the aligned sequences, such as glycine at positions 1 and 11, lysine at position 8, valine at position 9, and leucine at position 16. These amino acids are conserved in the five aligned sequences, suggesting that they are important for the development of their antimicrobial and antitumor functions.

#### 3.2.6. Chemical Characterization of Ramosin Peptide of *B. ramosi*

Table 4 shows the physicochemical properties of the Ramosin peptide. According to the properties obtained using a peptide property calculator (available online: https://www.novoprolabs.com/tools/calc_peptide_property, accessed on 15 November 2022), Ramosin is a peptide of 16 amino acids and has a net charge of +3, which classifies it as a cationic peptide, and has a grand average of hydropathicity index (GRAVY) of 0.92, indicating that it is a hydrophobic peptide [50].

Once its main physicochemical properties were determined, circular dichroism analysis was performed to determine the secondary structure of the peptide. In TFE at 30%, the spectrum shows a maximum before 200 nm and two minimums between 200 nm and 230 nm, which corresponds to an alpha-helical structure. This trend is preserved at all temperatures measured (Figure 9a). The CD spectra in water and other polar media showed unordered structure (random coil, Figure 9b).

Since the results of circular dichroism showed that the Ramosin peptide has a tendency to form an alpha helix, its structure was modeled using the I-TASSER server (available online: https://zhanggroup.org/I-TASSER/, accessed on 20 December 2020) [53]. The best model (with a C score of −0.44) shown in Figure 10a demonstrates the formation of an alpha helix, thus confirming the results obtained by circular dichroism.

Additionally, we used the NetWheels tool [55] to determine the amino acid distribution of the Ramosin peptide. It is observed that the nonpolar amino acids are located at one of the faces and the polar residues are located at the other face, confirming its amphipathic structure that is characteristic of AMPs (Figure 10b).

## 4. Discussion

To date, the peptides reported in salamanders have been identified through proteomic strategies, which require several individuals, and in some cases their sacrifice. In this work we proposed a methodology for the in silico search of peptides with potential antimicrobial activity. A strategy was followed based on the identification of peptides through alignments with the Fasta36 tool between the *B. ramosi* transcriptome and peptides reported in various databases [56].

Using the above strategy, a total of 91 candidate peptides were identified, of which 11 were selected to be chemically synthesized. Among these peptides is Ramosin, which corresponds to a unique sequence of *B. ramosi,* with no matches in the NCBI databases, when performing a BLAST search. Ramosin peptide has 16 amino acids, a net charge of +2, an alpha-helical secondary structure, and shows activity against Gram-negative and Gram-positive bacteria. It did not cause hemolysis of human erythrocytes up to a concentration of 128 μM and did not affect eukaryotic HaCat, MCF-7, and PC-3 cells.

Only two salamander peptides with antibacterial activity have been reported in the literature so far. Both peptides were identified in the salamander *Andrias davidianus* belonging to the Cryptobranchidae family. Andricin 01 consisting of 10 amino acids has a random structure and has anti-Gram-positive and anti-Gram-negative activity. In addition to this, it does not present hemolytic activity, nor does it affect eukaryotic cells [16]. Yiang et al. reported, in 2017, the first salamander cathelicidin with activity against Gram-positive and Gram-negative bacteria [57]. In 2000, Fredericks and Danker reported what would be the first peptide in pletodontids, such peptide showed activity against *S. aureus*, but not against *E. coli;* in addition, it also had hemolytic activity [58]. The other two peptides reported from salamanders were in *Ambystoma tigrinum* [59] and *Cynops fundigensis* [60]; both peptides showed activity against *S. aureus*. According to the above, the peptide Ramosin identified in *B. ramosi* would be the first peptide evaluated against three strains of Gram-positive bacteria and two strains of Gram-negative bacteria, showing activity against all of them and not causing hemolysis at a concentration up to four times its MIC.

Ramosin from *B. ramosi* has a high percentage of similarity with the Aureins, which are peptides identified in the frogs *Litoria aurea* and *Litoria raniformis* [49]. When an alignment of the aurein sequences 2.1, 2.2, 2.3, 2.4, 2.5, and 2.6 of *L. aurea* and *L. raniformis* and Ramosin peptide was made, it was found that positions 1, 8, 9, 11, and 16 are conserved in all Aureins and in Ramosin. The positions that presented the greatest variation were 12 and 13. This could suggest that the residues that are conserved, glycine (G) in positions 1 and 11, lysine (K) in position 8, valine (V) in position 9, and leucine (L) in position 16, may be important for the antimicrobial or antitumor activities of these peptides.

The Ramosin peptide from *B. ramosi* has a 62.5% identity and 93.8% similarity with Aureins 2.2 and 2.3 from the frog *Litoria aurea;* however, the concentrations at which they exert their antimicrobial activity varied. The Aureins of the frog *Litoria aurea* were evaluated only in Gram-positive bacteria. In *B. cereus,* for example, Aureins 2.2 and 2.3 presented an MIC of 62 μM, while with the Ramosin peptide it has an MIC of 35 μM. For *S. aureus,* a higher activity was observed with Aurein 2.2, which has an MIC of 15.5 μM; while Aurein 2.3 has an MIC of 62 μM and Ramosin peptide has an MIC of 20 μM [49]. It can be seen how, despite having a high percentage of similarity, the activity against Gram-positive bacteria is not the same; suggesting that in addition to the physicochemical properties of the amino acids, there are other factors that could determine the activity against this type of bacteria, being necessary to continue with the investigations about the mechanisms of action and the determination of essential amino acids for the activity of peptides, among other factors.

To verify if there are other factors that could affect the activity of the peptides, we compared the MICs of peptides reported in APD3 that had a +2 charge and a length between 16 and 20 amino acids, with the Ramosin peptide. A total of 76 peptides with these characteristics were obtained; subsequently, those that had been evaluated in one or more of the Gram-positive and Gram-negative bacteria used in this work were filtered, thus selecting 33 peptide sequences. We found that the Ramosin peptide exhibited higher activity against Gram-positive and Gram-negative bacteria than other peptides that have similar physicochemical properties. Such is the case of the Metalnikowin I peptide (AP00362) that has a similarity of 73.3% with the Ramosin peptide that does not present activity against *B. cereus* or against *E. faecalis* and has an MIC > 200 μM for *S. aureus* [34,61]; while the Ramosin peptide has an MIC of 35 μM for *B. cereus* and *E. faecalis* and 20 μM for *S. aureus.* Regarding Gram-negative bacteria, Metalnikowin I has an MIC > 200 μM for *P. aeruginosa* [34,61], while the Ramosin peptide presented an MIC of 17.5 μM for *E. coli* and *P. aeruginosa.* In general, the Ramosin peptide presented higher activity against Gram-negative bacteria when compared to other peptides that presented a net charge of +2 and a length between 16 and 20 amino acids reported in the APD3 database. The only peptide that showed higher activity than Ramosin was Phylloseptin-1 (AP00546) which has an MIC of 7.9 μM for *E. coli* and 4 μM for *P. aeruginosa* [33]. In addition, the peptide Bmkb1 (AP01977) had a higher activity than Ramosin against *E. coli* (9.5 μM) and *S. aureus* (8.4 μM), but a lower activity against *P. aeruginosa* (47.5 μM) when compared with the Ramosin peptide [34,62].

The above physicochemical properties, such as charge and length, are not sufficient to predict the possible antimicrobial activity of a peptide sequence since, as previously observed, peptides with the same charge and length vary in their activity against Gram-positive and Gram-negative bacteria. The percentage of hydrophobicity between Aureins 2.1, 2.2, 2.3, 2.4, 2.5, and 2.6 is 56%, which is close to that of the Ramosin peptide (50%), so hydrophobicity does not seem to be a variable that can explain the differences in activity. Lastly, we checked the secondary structure to see if they could partly explain the observed differences in activity. Aureins from the frog *Litoria aurea* have an alpha-helical secondary structure similar to the Ramosin peptide; therefore, it can be suggested that in addition to the physicochemical properties and the secondary structure of the peptide, there are other factors that could be determining the antimicrobial activity of peptides.

After analyzing the results of the antimicrobial activity assays, it was observed that the Ramosin peptide has a higher activity against Gram-negative than against Gram-positive bacteria. This could be due to the different membrane composition of both types of bacteria. The bacterial membrane is the main target of action of most cationic peptides, since they are negatively charged due to the presence of anionic lipids such as lipopolysaccharides (LPS) in Gram-negative bacteria or teichoic acids in Gram-positive bacteria. In Gram-negative bacteria, the interaction of cationic peptides with LPS results in perturbation of the outer membrane, which could explain the effect of the Ramosin peptide observed on the membrane of *E. coli* in scanning electron microscopy experiments. Scott et al. described the role of LPS in the activity of cationic peptides and mentioned that many of the peptides with reduced binding affinity for LPS also had decreased antimicrobial activity [63], suggesting that the Ramosin peptide might have binding affinity for LPS. In *S. aureus*, a disturbance in the outer membrane was also observed; this suggests that since Ramosin is a cationic peptide, it could interact with the negatively charged teichoic acid and thus exert its antimicrobial activity [64,65,66,67]. The results obtained by scanning electron microscopy suggest then that the main target of the Ramosin peptide could be the cytoplasmic membrane, since *S. aureus* exhibited bubble-like projections when treated with the peptide. This can result in severe membrane disruption, which could progressively lead to increased permeability and cell lysis [68].

According to the results discussed here, we consider that this work represents an important contribution related to peptides in salamanders, since there are very few publications in this area. Additionally, it is the first study of peptides carried out in an endemic species of salamander, and can become a reference for future research in this field. Despite the fact that the developed method allowed to identify a peptide sequence, Ramosin, that is capable of acting against Gram-negative and Gram-positive bacteria, we are aware that further work is needed to refine the in silico design of these molecules and to achieve optimal activity.

## Figures and Tables

**Figure 1 pharmaceutics-14-02579-f001:**
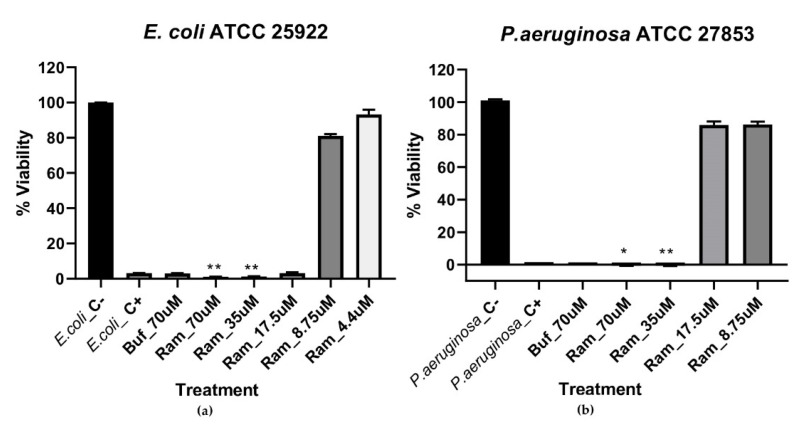
Endpoint assay to determine the viability percentage of Gram-negative bacteria after 19 h of treatment with Ramosin at concentrations of 4.37 μM, 8.75 μM, 17.5 μM, 35 μM, and 70 μM by broth microdilution assay. (**a**) *E. coli* (**b**) *P. aeruginosa*. Buf: Buforin II. C−: bacteria without treatment C+: bacteria treated with NaClO 0.012%. Absorbance reading at 600 nm. Three independent experiments were carried out for each bacterial strain. Data are presented as mean ± SD. *: *p* < 0.05, **: *p* < 0.01.

**Figure 2 pharmaceutics-14-02579-f002:**
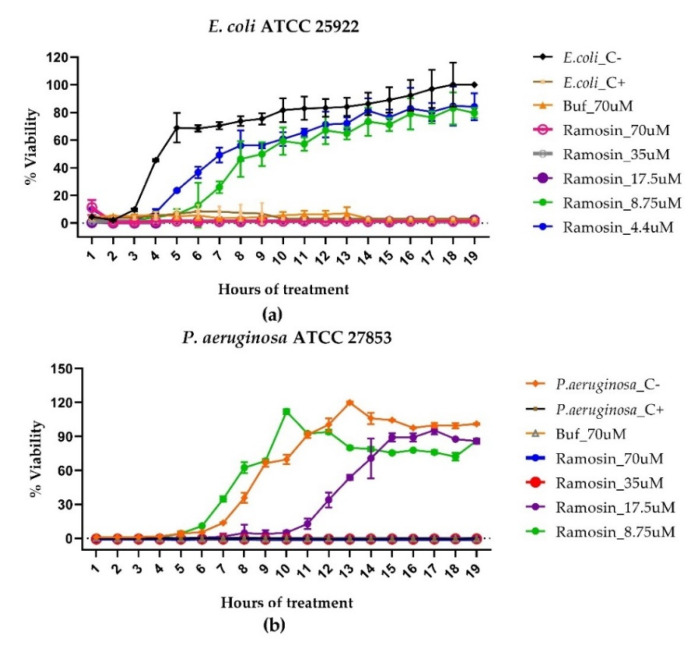
Growth inhibition kinetics of Gram-negative bacteria *E. coli* (**a**) and *P. aeruginosa* (**b**) treated with Ramosin peptide from *B. ramosi.* Absorbance reading at 600 nm every hour of treatment. Three independent experiments were carried out for each bacterial strain. Data are presented as mean ± SD. Buf: Buforin II. C−: bacteria without treatment C+: bacteria treated with NaClO 0.012%.

**Figure 3 pharmaceutics-14-02579-f003:**
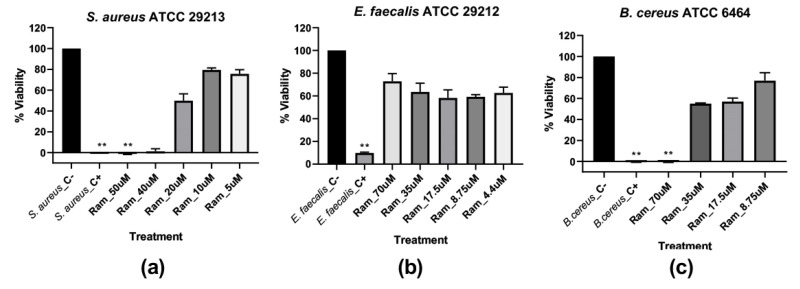
Endpoint assay to determine viability percentage of Gram-positive bacteria after 19 h of Ramosin treatment by broth microdilution assay. The bacteria *S. aureus*, *E. faecalis,* and *B. cereus* without treatment were used as a negative control, and NaClO at 0.012% was used as a positive control for death. (**a**) *S. aureus* ATCC 29213. (**b**) *E. faecalis* ATCC 29212. (**c**) *B. cereus* ATCC 6464. Absorbance reading at 600 nm. Three independent experiments were carried out for each bacterial strain. Data are presented as mean ± SD. **: *p* < 0.01.

**Figure 4 pharmaceutics-14-02579-f004:**
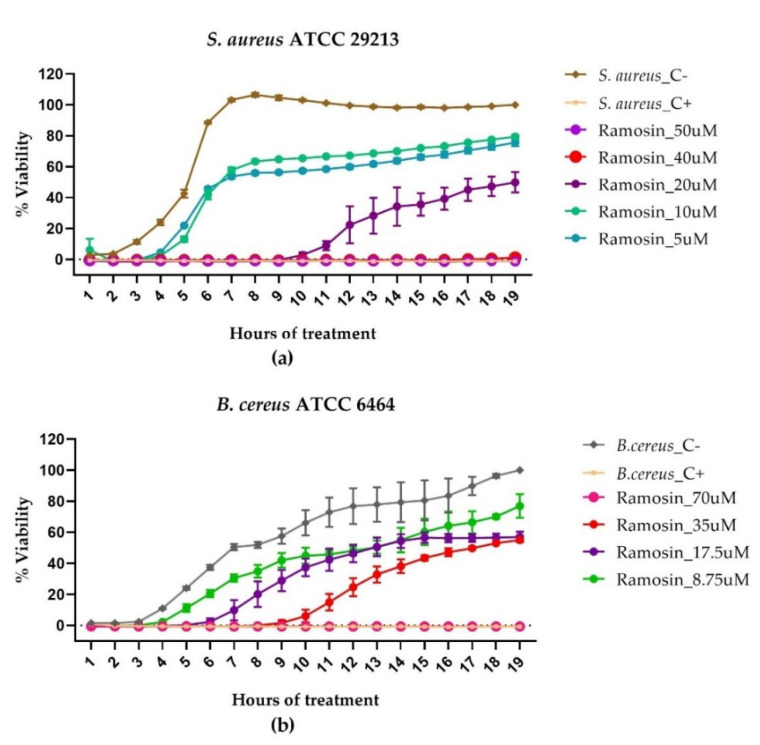
Growth inhibition kinetics of Gram-positive bacteria *S. aureus* and *B. cereus* treated with the Ramosin peptide from *B. ramosi.* Absorbance reading at 600 nm every hour of treatment. (**a**) *S. aureus.* (**b**) *B. cereus.* Three independent experiments were carried out for each bacterial strain. Data are presented as mean ± SD. C−: bacteria without treatment C+: bacteria treated with NaClO 0.012%.

**Figure 5 pharmaceutics-14-02579-f005:**
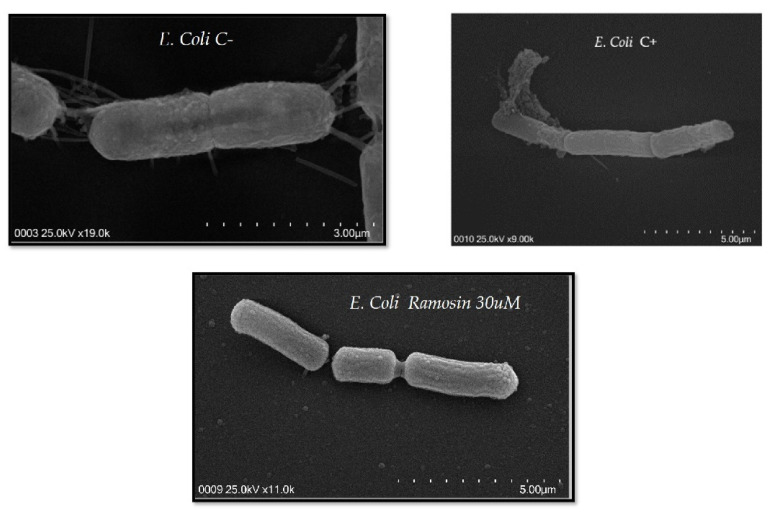
Scanning electron microscopy of the Gram-negative bacteria *E. coli*. Bacteria were treated with Ramosin peptide at 30 μM. C−: *E. coli* without treatment. C+: BTM-P1 peptide at 20 μM [47].

**Figure 6 pharmaceutics-14-02579-f006:**
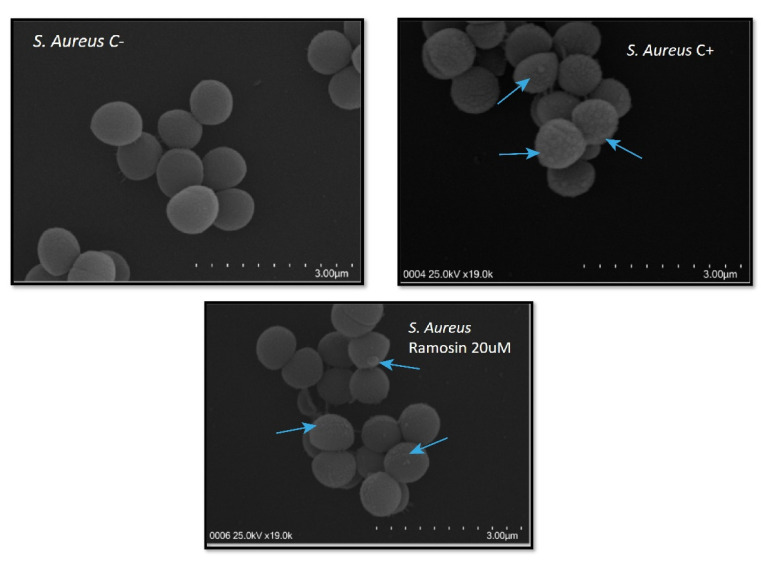
Scanning electron microscopy of the Gram-positive bacteria *S. aureus.* Bacteria were treated with Ramosin peptide at 20 μM. C−: *S. aureus* without treatment. C+: BTM-P1 peptide at 20 μM [47]. The blue arrows indicate the disturbances that occurred in the membrane of *S. aureus*.

**Figure 7 pharmaceutics-14-02579-f007:**
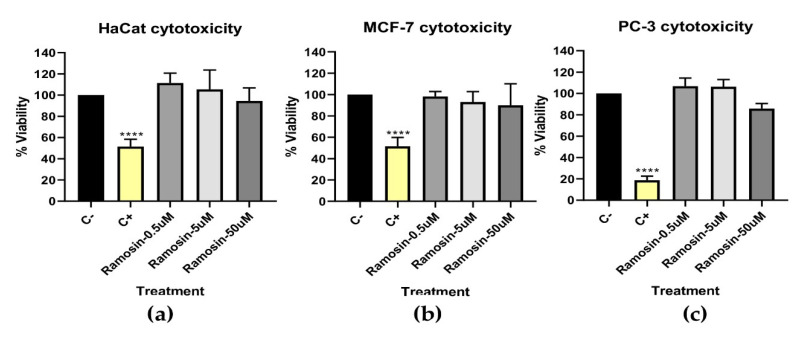
Screening test to determine the percentage of viability by the reduction of MTT in the cell lines (**a**) HaCat, (**b**) MCF-7, and (**c**) PC-3 treated with the Ramosin peptide at concentrations of 0.5 μM, 5 μM, and 50 μM for 24 h. C−: untreated cells. C+: cells treated with H_2_O_2_. Three independent experiments were carried out. Data are presented as mean ± SD. ****: *p* < 0.001.

**Figure 8 pharmaceutics-14-02579-f008:**
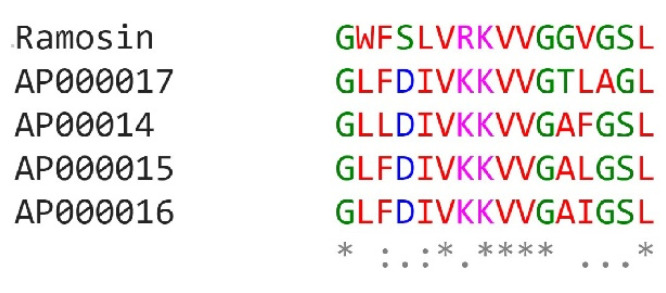
Alignment of the Ramosin peptide sequence from *Bolitoglossa ramosi* and the Aurein sequences reported in APD3: AP00017 (Aurein 2.4), AP00014 (Aurein 2.1), AP00016 (Aurein 2.3), AP00015 (Aurein 2.2) [49]. The asterisk (*) indicates positions which have a single, fully conserved residue. The colon (:) indicates conservation between groups of strongly similar properties and a period (.) indicates conservation between groups of weakly similar properties.

**Figure 9 pharmaceutics-14-02579-f009:**
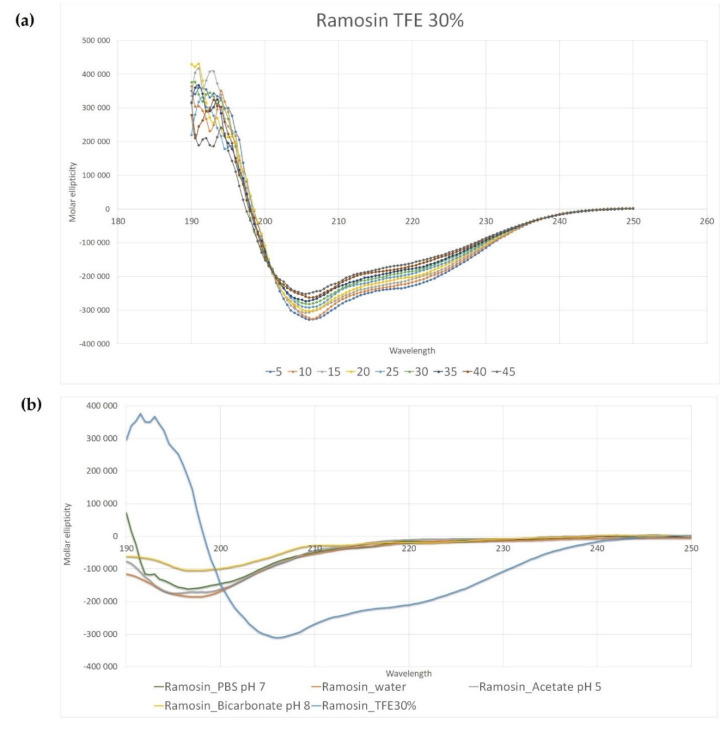
Circular dichroism analysis for the Ramosin peptide from *Bolitoglossa ramosi*. (**a**) CD spectra in TFE 30% at different temperatures. (**b**) Comparison of the spectra at different media and pH. TFE 30% is used to reduce the dielectric constant of the medium and favors intrachain interactions [51,52].

**Figure 10 pharmaceutics-14-02579-f010:**
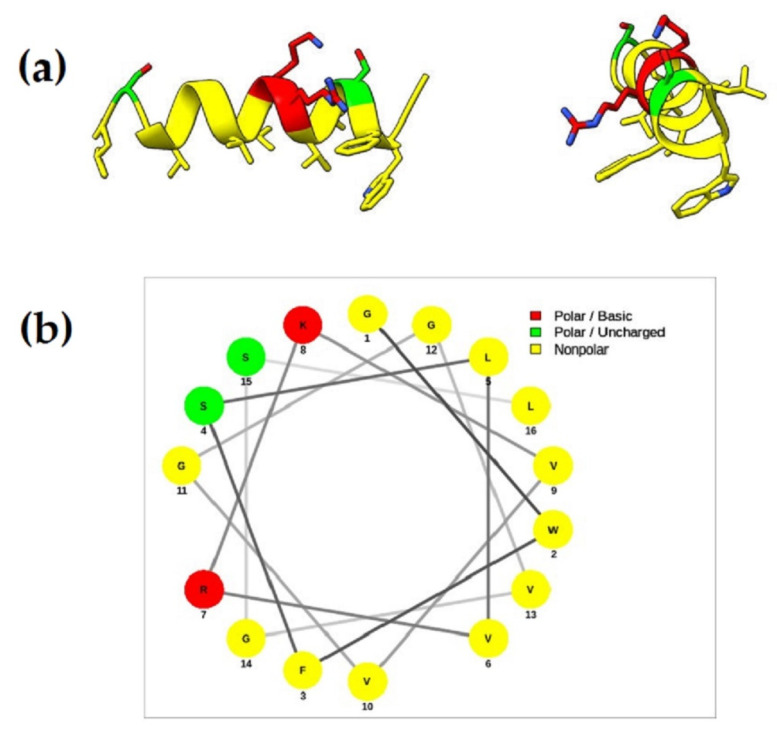
Ramosin peptide structure. (**a**) Tridimensional model obtained through I-TASSER server and built with Chimera X [54]. (**b**) Helical wheel representation. Positively charged polar amino acids are shown in red, uncharged polar amino acids in green, and nonpolar amino acids in yellow.

**Table 1 pharmaceutics-14-02579-t001:** Peptides reported in APD3 that appeared to be 100% identical to some hypothetical proteins of *B. ramosi*.

APD3	Length (aa*)	Net Charge	% Hydrophobicity	Peptide Source	Activity of the Reported Peptides on APD3 Database [34]
AP00176	30	+3	53	Neutrophils; natural killer cells, monocytes; saliva; *Homo sapiens*	Anti-Gram+ & Gram-, antiviral, antifungal, antiparasitic, anti-HIV, chemotactic, anti-**MRSA, anti-toxin, enzyme inhibitor, anti-sepsis, wound-healing, anticancer
AP00177	29	+3	51	Neutrophils; natural killer cells, monocytes; saliva; *Homo sapiens*	Anti-Gram+ & Gram-, antiviral, antifungal, anti-HIV, chemotactic, anti-toxin, enzyme inhibitor, anticancer
AP00208	32	+5	28	Pig, *Sus scrofa*	anti-Gram+
AP00308	21	+6	33	*Bufo bufo gargarizans.*	Anti-Gram+ & Gram-, antifungal, candidacidal, antiparasitic, anti-sepsis, anticancer
AP00336	12	0	50	Rainbow trout, *Oncorhynchus mykiss*	Anti-Gram+, antifungal
AP00449	13	+1	30	Brain, *Homo sapiens*	Anti-Gram+, antiviral, antifungal, candidacidal, anti-HIV
AP00528	7	−7	0	*Ovis aries*	Anti-Gram+ & Gram-
AP02017	31	+5	45	Placental tissue, *Homo sapiens*	Antifungal, candidacidal
AP02030	74	0	33	Gill, Pacific oyster, *Crassostrea gigas*	Anti-Gram+ & Gram-
AP02096	59	+19	25	*Homo sapiens;* cytosol, macrophage, RAW264.7, *Mus musculus; Rattus norvegicus*	Anti-Gram+ & Gram-, anti-MRSA
AP02231	19	+2	26	Corneas, eyes, *Homo sapiens*	Anti-Gram-
AP02257	130	+8	40	Secretions and tissues, tears, saliva, human milk, and mucus; *Homo sapiens*	Anti-Gram+ & Gram-, antifungal, synergistic AMPs
AP02343	99	−2	31	Human amniotic fluid, *Homo sapiens*	Anti-Gram+ & Gram-
AP02441	6	0	16	*Streptomyces amritsarensis*	Anti-Gram+, anti-MRSA
AP02791	22	+7	22	Blood plasma, *Varanus* *komodoensis*	Anti-Gram+ & Gram-
AP02807	103	+18	33	American cupped oysters, *Crassostrea virginica*	Anti-Gram-
AP02813	24	+3	41	Leukocytes; the Russian Sturgeon, *Acipenser gueldenstaedtii*	Anti-Gram-
AP02884	7	−6	0	*Ovis aries*	Anti-Gram-
AP02885	7	−5	14	*Ovis aries*	Anti-Gram-
AP02984	6	−1	33	Bovine milk digestion by bacteria, *Bos taurus*	Anti-Gram+
AP03140	13	0	53	Excretions and Secretions, *Sarconesiopsis magellanica*	Anti-Gram+ & Gram-
AP03151	20	+7	35	Fragments of a human antimicrobial protein	Anti-Gram+ & Gram-, antifungal, candidacidal, anti-inflammatory, anti-sepsis
AP03159	20	+1	50	Endometrial fluid peptides, *Homo sapiens*	Anti-Gram+ & Gram-, antifungal, anti-MRSA

aa*: amino acids. **MRSA: methicillin-resistant *Staphylococcus aureus.*

**Table 2 pharmaceutics-14-02579-t002:** Candidate peptides of *B. ramosi* chemically synthesized according to the similarity found with peptides in APD3 database.

Candidate Peptide Code	Peptide Sequences Synthesized	Net Charge at pH 7	MW (g/mol)	Family	APD3 Similar Peptide	(%) Similarity with APD3 Peptides *
3412	GWFSLVRKVVGGVGSL-NH2	+3	1659.96	Aurein	AP00016	93.8
3413	KQYQLVERIIGSIGSL-NH2	+2	1803.1	Aurein	AP00016	87.5
3414	GMLMMVRRPFGPFGSI-NH2	+3	1795.24	Aurein	AP00014	87.5
3415	QLHDVMKRVAKSF-NH2	+3	1557.85	Aurein	AP00013	84.6
3416	GAFDDVKKVATTI-NH2	+1	1363.55	Aurein	AP00012	80
3417	SLLSAWGKILGSKLNEKLTQ-NH2	+3	2185.55	Dahlein	AP00703	80.1
3418	GFLNYYRRFIGSFAEVVT-NH2	+2	2138.42	Maculatin	AP00262	83.3
3419	VLSPSLGSLAGVLGGVLKLA-NH2	+2	1850.24	Maximin	AP00832	85
3420	RVRLEACVRGICRRNCK-NH2	+6	2031.48	Tachyplesin	AP00214	80.3
3424	GIFTLIHCSLEGKVKKIECS-NH2	+2	2204.64	Odorranain-J1	AP01298	80
3428	AVAGRSQGQ-NH2	+2	871.94	Cn-AMP1	AP01342	100

* The similarity was calculated with Fasta36 alignment of the transcript with the APD3 database.

**Table 3 pharmaceutics-14-02579-t003:** Number of reads of the Ramosin peptide. Each of the samples used for de novo assembly of the *B. ramosi* transcriptome is shown. Control: limb samples at the time of amputation, Limb: tissue after regeneration, Blastema: tissue in the process of regeneration.

Sample	Code	Number of reads	% Discordance
Control	BRE001	6	1.32
	BRE002	4	0.248
	BRE003	22	5.176
Limb	BR006	18	4.125
	BR007	18	0.495
Blastema	BR004	8	0.866
	BR005	2	0.495
	BR008	10	0.693
	BR009	0	0
	BR010	4	0.99
Intestine	BRI001	20	1.139
Skin	BRP001	2	1.98

**Table 4 pharmaceutics-14-02579-t004:** Physicochemical properties of Ramosin peptide from *B. ramosi*.

Physicochemical Properties of Ramosin Peptide	
Sequence of amino acids	GWFSLVRKVVGGVGSL-NH2
Number of amino acids	16
Molecular weight	1659.96
Chemical formula	C_78_H_126_N_22_O_18_
Isoelectric point	11.48
Net charge	+3
GRAVY	0.92

## Data Availability

Not applicable.

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
