# Peer review of "Ramosin: The First Antibacterial Peptide Identified on Bolitoglossa ramosi Colombian Salamander"

_pharmaceutics, 2022, doi:10.3390/pharmaceutics14122579_

Round 1

Reviewer 1 Report

The paper fit with aim and scopes of the journal Pharmaceuticals. However, I have the following major remarks to the manuscript:

1. First of all English language of the paper has to be corrected by Native English speaker. There are many places where it is not correct and comprehensible. For example:

- line 38 and 39 there are two times directly

- line 40 lesser has to be less

- line 87 there are two times data

- line 104 NO has to be replaced by lack of, etc.

2. In the Abstract section authors abbreviate antimicrobial peptides with AMP. Further often they do not use abbreviation and why in this case they make this abbreviation if they do not use? Secondly one time they abbreviate this term with AMPs and another with AMP. Please chose one and use it in the whole manuscript. Moreover, may be authors are not familiar but first the abbreviation is defined in the text and secondly the abbreviation is presented in brackets. NOT in opposite. See line 69 (m.a.s.l) and finally why one term has to be abbreviated if it is not used another time in the manuscript? What is the aim of this abbreviation?

3. The manuscript start directly with figure 2 in the introduction section (line 62), and figure 1 comes in the Results section (line 283). Generally the order is 1, 2, 3, NOT 1,5,2,7, etc.

4. I can see that authors are absolutely not familiar with peptide synthesis. My impression is that even they are not work on this field. The synthesis of peptides are presented in a very simplified manner. For synthesis of aim peptides they cite an article number 40 but it is not enough. Please give the methodology for peptide synthesis even in a short description. Nobody will go to another article to search for this information. In addition, why authors abbreviate system for peptide cleavage from the resin (line 166) and further they give the full names of reagent? What is the aim? The same is in line 187, 189, etc. What authors mean in line 547 that peptide are synthesized in a solid phase? Does that mean using solid phase peptide synthesis approach for synthesis on solid support (resin) or really they mean in a solid phase which mean without presence of solvent in a reaction mixture?

5. Authors describe that they use flash chromatography to purify the peptide by means of gradient of solvents (lines 167-169). But the information is not enough, for how many time they make this gradient? How the gradient is constructed in time?

6. There is no any data about the CD apparatus (line 173-176).

7. In the experimental part there is no any information about the concentration range where peptide are tested for their antimicrobial properties. We can see different concentrations only in presented figures. Please define the concentration range of all experiments.

8. The generalities for descriptions of bacterial strains in Italic are not respected.

9. Results presented in figures 3a and b are incomprehensible. I can see 8 described curves in the legend of 3a and 7 for 3b and I saw 6 lines in the first one and 4 in the second. So may be some curves match but the manner of presentation do not give possibility to understand. The same is for figure 5.

10. Many sentences are presented separated with ; Why? (see lines 577 to 589). These are separated sentences. What is this manner to separate them not with a full stop but with ; ?

Finally my opinion is that this article need of major revision in order to be evaluated for publication in the journal Pharmaceuticals.

Author Response

First of all, we want to thank the reviewers for their enriching comments. We have carefully reviewed the manuscript and believe we have responded to all requests. Below we present the detailed responses for each reviewer.

Reviewer 1

Comments and Suggestions for Authors

The paper fit with aim and scopes of the journal Pharmaceuticals. However, I have the following major remarks to the manuscript:

  1. First of all English language of the paper has to be corrected by Native English speaker. There are many places where it is not correct and comprehensible. For example:

- line 38 and 39 there are two times directly

- line 40 lesser has to be less

- line 87 there are two times data

- line 104 NO has to be replaced by lack of, etc:

The manuscript was reviewed and the errors mentioned in lines 38,39,40, 87 and 104 were corrected.

  1. In the Abstract section authors abbreviate antimicrobial peptides with AMP. Further often they do not use abbreviation and why in this case they make this abbreviation if they do not use? Secondly one time they abbreviate this term with AMPs and another with AMP. Please chose one and use it in the whole manuscript. Moreover, may be authors are not familiar but first the abbreviation is defined in the text and secondly the abbreviation is presented in brackets. NOT in opposite. See line 69 (m.a.s.l) and finally why one term has to be abbreviated if it is not used another time in the manuscript? What is the aim of this abbreviation?

Thanks for the warning, the abbreviations were revised and the style unified taking into account the suggestions of the reviewer.

  1. The manuscript start directly with figure 2 in the introduction section (line 62), and figure 1 comes in the Results section (line 283). Generally the order is 1, 2, 3, NOT 1,5,2,7, etc.

Apologies for the error, we have revised and fixed the numbering of tables and figures within the text.

  1. I can see that authors are absolutely not familiar with peptide synthesis. My impression is that even they are not work on this field. The synthesis of peptides are presented in a very simplified manner. For synthesis of aim peptides they cite an article number 40 but it is not enough. Please give the methodology for peptide synthesis even in a short description. Nobody will go to another article to search for this information. In addition, why authors abbreviate system for peptide cleavage from the resin (line 166) and further they give the full names of reagent? What is the aim? The same is in line 187, 189, etc. What authors mean in line 547 that peptide are synthesized in a solid phase? Does that mean using solid phase peptide synthesis approach for synthesis on solid support (resin) or really they mean in a solid phase which mean without presence of solvent in a reaction mixture?

We apologize for this error; Section 2.2 corresponding to peptide synthesis has been rewritten in more detail, in addition to including the relevant references.

  1. Authors describe that they use flash chromatography to purify the peptide by means of gradient of solvents (lines 167-169). But the information is not enough, for how many time they make this gradient? How the gradient is constructed in time?

The peptides were analyzed by Reverse Phase High Performance Liquid Chromatography (RP-HPLC), using a gradient of acetonitrile of 0-70%, this was included in the text.

  1. There is no any data about the CD apparatus (line 173-176).

The information of the equipment is now included in the text.

  1. In the experimental part there is no any information about the concentration range where peptide are tested for their antimicrobial properties. We can see different concentrations only in presented figures. Please define the concentration range of all experiments.

The reviewer's suggestion was accepted and the information about the peptide concentration used was specified in each experiment (lines 226 and 227, line 240, line 252 and line 274).

  1. The generalities for descriptions of bacterial strains in Italic are not respected.

Sorry for the lapse, it was an error at some point when updating the format. We have revised and included italics where appropriate.

  1. Results presented in figures 3a and b are incomprehensible. I can see 8 described curves in the legend of 3a and 7 for 3b and I saw 6 lines in the first one and 4 in the second. So may be some curves match but the manner of presentation do not give possibility to understand. The same is for figure 5.

The reviewer's suggestion was accepted and Figures 3 (now Figure 2) and 5 (now Figure 4) were modified to try to improve the understanding of the curves that are superimposed. Figure 2 is on line 415, and Figure 4 is on line 445.

  1. Many sentences are presented separated with ; Why? (see lines 577 to 589). These are separated sentences. What is this manner to separate them not with a full stop but with ; ?

The paragraph was corrected.

Finally my opinion is that this article need of major revision in order to be evaluated for publication in the journal Pharmaceuticals.

The manuscript was reviewed by a native speaker, we hope we have corrected all errors.

Reviewer 2 Report

This work was focused on identifying antimicrobial peptides from the endemic Colombian salamander B. ramosi transcriptome sequences using bioinformatics. Peptide 3412 was identified as Ramosin, and its activity and hemolytic activity were assessed. This work will fetch interest among researchers in AMPs research area.

In general, this manuscript is well written, but some points listed below should be properly addressed before this MS could be recommended for publication:

Please rewritten some part of manuscript briefly because of repetition, such as line 323 and line 347-349.

Line 62,  (Figure 2) ?

Figure 1 is recommended to be deleted, because of repetition with text.

Table 2, please provide Net charge, measured MS of 11 peptides, Identity (%) with peptide in APD3 database.

Please provide the HPLC chromatogram and MS spectrum of synthesized peptides as supplementary file.

Name of microorganisms should be typed in italic, such as E. coli and S.typhimurium.

Please provide the negative and positive control in Fig 5. Also do statistical analysis for different treatment at 19 h in Fig 3 and Fig 5.

Line 416, peptide 3412 ?

Fig 7 and Fig 8, BTM-P1 peptide concentration?

Line 448-453, Results of mitochondrial membrane potential and cell membrane integrity and apoptosis is very interesting, please provide.

Fig 10, this Fig is not easily to see, fig quality is poor.

Fig 11, b is another formation of a result? If yes, keep one.

Line 505, GRAVY, 0.019 should be 0.919?

Author Response

First of all, we want to thank the reviewers for their enriching comments. We have carefully reviewed the manuscript and believe we have responded to all requests. Below we present the detailed responses for each reviewer.

Reviewer 2

Comments and Suggestions for Authors

This work was focused on identifying antimicrobial peptides from the endemic Colombian salamander B. ramosi transcriptome sequences using bioinformatics. Peptide 3412 was identified as Ramosin, and its activity and hemolytic activity were assessed. This work will fetch interest among researchers in AMPs research area.

In general, this manuscript is well written, but some points listed below should be properly addressed before this MS could be recommended for publication:

Please rewritten some part of manuscript briefly because of repetition, such as line 323 and line 347-349.

Thanks for the call of attention, we have reviewed the manuscript and corrected these errors. (Line 323, now line 362, was maintain, but lines 347-349 were eliminated.

Line 62,  (Figure 2) ?

Sorry for the typing error, it was removed from the text. All tables and figures have been revised and numbered correctly.

Figure 1 is recommended to be deleted, because of repetition with text.

Thanks for the suggestion, Figure 1 was removed from the text.

Table 2, please provide Net charge, measured MS of 11 peptides, Identity (%) with peptide in APD3 database.

The reviewer's suggestion was accepted and Table 2 (line 332) was modified adding the requested information. Net charge was provided for each peptide synthesized. The MALDI and ESI results showed that the masses of the synthesized peptides match the expected mass. In the Figure S4 were provided all the HPLC and ESI-MS of the 11 peptides synthesized. The comparison with the database gives data of % of similarity not of identity, then this is included in the table.

Please provide the HPLC chromatogram and MS spectrum of synthesized peptides as supplementary file.

HPLC chromatograms and MS spectrums of synthesized peptides were added on supplementary file as Figure S4.

Name of microorganisms should be typed in italic, such as E. coli and S. typhimurium.

Sorry for the lapse, it was an error at some point when updating the format. We have revised and included italics where appropriate.

Please provide the negative and positive control in Fig 5. Also do statistical analysis for different treatment at 19 h in Fig 3 and Fig 5.

The information about positive and negative controls are now included in Figure 2 (now and Figure 4 (Fig 3 and 5 before). Also, the statistical analysis was made for different treatment at 19 h in both figures.

Line 416, peptide 3412 ?

Apologize for the lapse, 3412 was changed for Ramosin (line 453).

Fig 7 and Fig 8, BTM-P1 peptide concentration?

The concentration of BTM-P1 peptide was included in Figure 5 and Figure 6 (Figure 7 and 8 before).

Line 448-453, Results of mitochondrial membrane potential and cell membrane integrity and apoptosis is very interesting, please provide.

Thanks for the suggestion, the results of flow cytometry to evaluate mitochondrial membrane potential with DIOC6 and PI as well as the information related with membrane integrity and apoptosis are now included in supplementary information (Figure S6 and Fig S7 respectively).

Fig 10, this Fig is not easily to see, fig quality is poor.

The quality of the figure was improved, and the Figure 10 was moved to supplementary information for suggestion of reviewer 3.

Fig 11, b is another formation of a result? If yes, keep one.

The reviewer's suggestion was accepted and Figure 8 (Figure 11b before, line 524) was deleted, because in Figure 11a it is possible to observe the amino acids conserved between the different peptides.

Line 505, GRAVY, 0.019 should be 0.919?

Sorry for the typo, it was corrected.

Reviewer 3 Report

The work contains interesting results, but I have two significant remarks regarding the presented results:

#1

Figure 2 shows the results of the ramosine antibiotic activity test. As for E.coli bacteria, at a concentration greater than 17.5 microM, the peptide kills the bacterium. However, if we look at Suppelmental Data Figure 3S, then for the same bacteria the concentration of 30 microM does not stop its growth. The Materials and Methods section describes only one method for testing antibacterial activity, so I assume that the results in Figure 2 and Figure 3SA are obtained under the same conditions. This is not a problem with the follow-up time either, as the data shown in Figure 3 indicate that ramosine works immediately after administration. How the authors explain the significant discrepancies between the results shown in Figure 2 and Figure 3S. The same discrepancy can also be seen in the case of the results for Bufforine II.

#2

In Materials and Methods, the procedure for chemical synthesis of peptides is described: “Candidate peptides were synthesized by solid phase multiple peptide system [40] 164 using Fmoc amino acids (Iris and Rink resin 0.55 meq / g). - page 4 lines 164-165 "

From the brief description, I conclude that the reagents for the synthesis of peptides were obtained by the authors from Iris Biotech GmbH. A solid Rink support was used for the peptide synthesis. The company Iris offers resins of the Rink type, however, they are resins used for the synthesis of peptides with a C-terminal amide group (Rink-amide). The company Iris offers a different type of resins for the synthesis of peptides with a free carboxyl group. However, according to the brief description and the protocol used to cleave the peptide from the support, it can be virtually assured that the peptides presented in the work contain a C-terminal amide moiety. This is also confirmed by the data in Table 2, assuming that the weights are derived from MALDI-MS measurements. The rejected AVAGRSQGQ peptide has the correct mass of 872.8 Da, which corresponds theoretically to a mass of 872.45 Da for the peptide with the C-terminal amide group plus the mass of the proton as in the MALDI measurements. The peptide with a free carboxyl group should have a mass of 873.44 Da (peptide mass + proton mass) from MALDI measurements. Summing up, it seems that the authors are not fully aware of exactly what peptides they are examining. Thus, the data given in Table 4 characterizing the ramosine peptide is not true and should be corrected.

The work is extremely long and it could be significantly shortened and simplified if some of the results were transferred to Supplemental Data.

Section 3.2.2 may be transferred in its entirety to Supplemental Data. Studies have been carried out but the results show no hemolytic activity.

The results from Sction 3.2.3  microscopic observations do not bring new knowledge about the mechanism of action of the peptide, so these data can be transferred to Supp.Data and referred to in the discussion.

Similary Section r is 3.2.4.

Figure 12. Panel A can be removed, the same data are also in part B. It would be much more interesting to show the CD spectrum of the peptide in pure water or buffer solution. The use of TFE forces a specific structure, and how does the peptide behave without the addition of TFE, e.g. for different pH values?

Figure 13. Figure description. These are tertiary and not secondary structures. What are these descriptions under each model? Images of the structures of individual models alone will suffice. Panel B does not add any new information and can be removed.

Chapter 3.2.5 Should contain a link to work 23. Figure 10 can be transferred to Suppelemtal Data.

Literature is cited in many cases very carelessly and chaotically, which makes it difficult to find or sometimes impossible to identify (only some examples are provided)

[18] T. Lüddecke, “A salamander ’ s toxic arsenal : review of skin poison diversity and function in true salamanders , 719 genus Salamandra,” 2018.

[21] S. M. Rovito, G. Parra-olea, C. R. Vásquez-almazán, R. Luna-reyes, and D. B. Wake, “Deep divergences and 725 extensive phylogeographic structure in a clade of lowland tropical salamanders,” 2012

[31] A. Bin Hafeez, X. Jiang, and P. J. Bergen, “Antimicrobial Peptides : An Update on Classifications and Databases,” 745 2021.

 The entire manuscript should be read and verified by native speaker.

Author Response

First of all, we want to thank the reviewers for their enriching comments. We have carefully reviewed the manuscript and believe we have responded to all requests. Below we present the detailed responses for each reviewer.

Reviewer 3

Comments and Suggestions for Authors

The work contains interesting results, but I have two significant remarks regarding the presented results:

#1

Figure 2 shows the results of the ramosine antibiotic activity test. As for E.coli bacteria, at a concentration greater than 17.5 microM, the peptide kills the bacterium. However, if we look at Suppelmental Data Figure 3S, then for the same bacteria the concentration of 30 microM does not stop its growth. The Materials and Methods section describes only one method for testing antibacterial activity, so I assume that the results in Figure 2 and Figure 3SA are obtained under the same conditions. This is not a problem with the follow-up time either, as the data shown in Figure 3 indicate that ramosine works immediately after administration. How the authors explain the significant discrepancies between the results shown in Figure 2 and Figure 3S. The same discrepancy can also be seen in the case of the results for Bufforine II.

The protocols used for the tests were different. In the case of Figure 2, the protocol described in the material and methods section was used, which is a quantitative protocol, while the protocol in Appendix A was used for the screening test. In the latter case, the evaluation of viability is calculated based on the number of cells in the plate according to the dilutions, considering that a concentration of 1x107 is used in this context the difference between 17.5 and 30 μM corresponds approximately to one dilution (or one logarithmic unit). This is an approximate method used as a first filter to assess potential activity. Buforin II was not used in the screening method, instead the peptide BTM-P1 was used as C+.

Protocol for the screening test (Appendix A)

The screening assay is a rapid test to determine which candidate peptides have antibacterial activity. This test allows to determine the activity of the peptides by means of the assay in 96-well plates in which each peptide is evaluated in triplicate. To perform the screening test, bacteria are grown in 5mL of liquid TSB medium overnight. The next day, 100 µL of this culture are taken and added to 5 mL of new TSB. The bacteria are allowed to grow under agitation at 37°C until they reach an OD between 0.2 and 0.4 to ensure that they are in the exponential phase of growth. The density of bacteria is adjusted to 1x107 bacteria (An OD of 0.001 is equivalent to 1x107 bacteria) and 10μL of the peptides are added to the 96-well plate at an intermediate concentration of 30 μM along with 90 uL of the bacteria suspension adjusted to 1x107. The 96-well plates were incubated for 24 h at 37 °C without shaking. Each peptide was tested in triplicate in a single assay. Once the incubation time was over, the plates were visually inspected to determine the dilution where the bacteria did not grow and the viability percentage of the bacteria was determined, taking 1x107 as 100%.

#2

In Materials and Methods, the procedure for chemical synthesis of peptides is described: “Candidate peptides were synthesized by solid phase multiple peptide system [40] 164 using Fmoc amino acids (Iris and Rink resin 0.55 meq / g). - page 4 lines 164-165 " From the brief description, I conclude that the reagents for the synthesis of peptides were obtained by the authors from Iris Biotech GmbH. A solid Rink support was used for the peptide synthesis. The company Iris offers resins of the Rink type, however, they are resins used for the synthesis of peptides with a C-terminal amide group (Rink-amide). The company Iris offers a different type of resins for the synthesis of peptides with a free carboxyl group. However, according to the brief description and the protocol used to cleave the peptide from the support, it can be virtually assured that the peptides presented in the work contain a C-terminal amide moiety. This is also confirmed by the data in Table 2, assuming that the weights are derived from MALDI-MS measurements. The rejected AVAGRSQGQ peptide has the correct mass of 872.8 Da, which corresponds theoretically to a mass of 872.45 Da for the peptide with the C-terminal amide group plus the mass of the proton as in the MALDI measurements. The peptide with a free carboxyl group should have a mass of 873.44 Da (peptide mass + proton mass) from MALDI measurements. Summing up, it seems that the authors are not fully aware of exactly what peptides they are examining. Thus, the data given in Table 4 characterizing the ramosine peptide is not true and should be corrected.

Section 2.2 corresponding to peptide synthesis has been rewritten in more detail, in addition to including the relevant references. The complete information of ESI-MS of the 11 peptides synthesized was provided on Figure S4. In Figure S4.11 is presented the ESI-MS of peptide 3428, the theoretical ions expected were: 873.8, 437.4, 291.9, and 219.2. Instead, we observed ions of 1094.4 and 548.0. That is why peptide 3428 was excluded from the experiments.

Table 4 was not modified, because the data presented there correspond to the theoretical mass predicted with the Expasy ProtParam tool (available online: https://web.expasy.org/protparam/) and it is not related with the results obtained with the ESI-MS, however, the observed ions correspond to the predicted theoretical weight.

The work is extremely long and it could be significantly shortened and simplified if some of the results were transferred to Supplemental Data.

Section 3.2.2 may be transferred in its entirety to Supplemental Data. Studies have been carried out but the results show no hemolytic activity.

Thanks for the suggestion. Figure 6 was changed to supplementary information; however, we consider that section 3.2.2 should go in the main text due to the little information that there is about AMPs identified in salamanders (less than 10). In addition, we consider it important to show complete information on hemolysis because some of the peptides reported in salamanders are hemolytic, as can be seen in:

  • R. Fredericks, L. P., & Dankert, “Antibacterial and hemolytic activity of the skin of the terrestrial salamander, Plethodon cinereus,” J. Exp. Zool., vol. 287, no. 5, pp. 340–345, 2000, Accessed: Dec. 27, 2020. [Online]. Available: https://pubmed.ncbi.nlm.nih.gov/10980492/.
  • Meng, S. Yang, C. Shen, K. Jiang, M. Rong, and R. Lai, “The First Salamander Defensin Antimicrobial Peptide,” PLoS One, vol. 8, no. 12, p. e83044, Dec. 2013, doi: 10.1371/journal.pone.0083044.

The results from Section 3.2.3  microscopic observations do not bring new knowledge about the mechanism of action of the peptide, so these data can be transferred to Supp.Data and referred to in the discussion

The direct effect of the peptide on the membrane is being evidenced, and it is also not common to present microscopy data in these cases, so considering that it is a new peptide and that the results clearly show the differences in activity on the two bacteria, we think that the section deserves to be in the main text.

Similarly Section r is 3.2.4.

Thanks for the suggestion, however again due to the little information on salamander AMPs, and considering a potential use in humans, it is important to consider that there is no cytotoxic effect on eukaryotic cells. Additionally, this essay has been reported in other publications such as:

  • Yang, B. Lu, D. Zhou, L. Zhao, W. Song, and L. Wang, “Identification of the first cathelicidin gene from skin of Chinese giant salamanders Andrias davidianus with its potent antimicrobial activity,” Dev. Comp. Immunol., vol. 77, pp. 141–149, Dec. 2017, doi: 10.1016/j.dci.2017.08.002.
  • Pei and L. Jiang, “Antimicrobial peptide from mucus of Andrias davidianus: screening and purification by magnetic cell membrane separation technique,” Int. J. Antimicrob. Agents, vol. 50, no. 1, pp. 41–46, Jul. 2017, doi: 10.1016/j.ijantimicag.2017.02.013.

Figure 12. Panel A can be removed, the same data are also in part B. It would be much more interesting to show the CD spectrum of the peptide in pure water or buffer solution. The use of TFE forces a specific structure, and how does the peptide behave without the addition of TFE, e.g. for different pH values?

Ramosin peptide CD spectra were performed in different media and at different pH, and are presented in Figure 9. The use of TFE 30% is a common practice in measuring CD spectra of peptides, and this has been reported (see Ref 52), the effect of TFE is not to force the formation of a structure, but by decreasing the dielectric constant, it favors intrachain interactions, which allows stabilizing the structures that present these interactions, as is the case of alpha helices.

Figure 13. Figure description. These are tertiary and not secondary structures. What are these descriptions under each model? Images of the structures of individual models alone will suffice. Panel B does not add any new information and can be removed.

We modified the figure and included only the best model (first model obtained in the server) and compared with the helical wheel representation.

Chapter 3.2.5 Should contain a link to work 23.

The reference was included.

Figure 10 can be transferred to Suppelemtal Data.

The reviewer's suggestion was accepted and the quality of the figure was improved. Figure 10 was moved to supplementary information.

Literature is cited in many cases very carelessly and chaotically, which makes it difficult to find or sometimes impossible to identify (only some examples are provided)

[18] T. Lüddecke, “A salamander ’ s toxic arsenal : review of skin poison diversity and function in true salamanders , 719 genus Salamandra,” 2018.

[21] S. M. Rovito, G. Parra-olea, C. R. Vásquez-almazán, R. Luna-reyes, and D. B. Wake, “Deep divergences and 725 extensive phylogeographic structure in a clade of lowland tropical salamanders,” 2012

[31] A. Bin Hafeez, X. Jiang, and P. J. Bergen, “Antimicrobial Peptides: An Update on Classifications and Databases,” 745 2021.

The references were reviewed, and now are in the correct format.

The entire manuscript should be read and verified by native speaker.

The manuscript was reviewed by a native speaker, we hope we have corrected all errors.

Round 2

Reviewer 1 Report

I accept the corrections made by the authors

Author Response

We want to thank you for their enriching comments.

Reviewer 3 Report

I thank the authors for the revised version of the manuscript and the corrections made.

However, I have one important point. It is clear from the revised version of the manuscript (description of the peptide synthesis and solid support used, and mass spectra) that all synthesized peptides contain a C-terminal amide group. For this reason, the data in Table 2 are incorrect. The peptide sequences should be presented in the format (for example, for sequence 3412) GWFSLVRKVVGGVGSL-NH2. The designation of NH2 according to the IUPAC nomenclature at the C-terminus of the peptide indicates the presence of an amide group, or the table description should state that all peptides contain a C-terminal amide group. Since peptides contain an amide group at the C-terminus, the formal pH charges should be corrected (+1 value added) e.g. peptide 3412 should have a charge of +3, peptide 3413 +2, peptide 314 +3 and so on. Table 4 should also be corrected, because the authors have never tested the activity of the GWFSLVRKVVGGVGSL peptide and the GWFSLVRKVVGGVGSL-NH2 peptide. So either this Table should be deleted or the values given there corrected (molar mass, chemical formula, predicted isoelectric point, GRAVY index and stability index). The authors do not seem to understand the difference between a peptide with a free carboxyl group and a peptide with a C-terminal carboxamide group.

Author Response

However, I have one important point. It is clear from the revised version of the manuscript (description of the peptide synthesis and solid support used, and mass spectra) that all synthesized peptides contain a C-terminal amide group. For this reason, the data in Table 2 are incorrect. The peptide sequences should be presented in the format (for example, for sequence 3412) GWFSLVRKVVGGVGSL-NH2. The designation of NH2 according to the IUPAC nomenclature at the C-terminus of the peptide indicates the presence of an amide group, or the table description should state that all peptides contain a C-terminal amide group. Since peptides contain an amide group at the C-terminus, the formal pH charges should be corrected (+1 value added) e.g. peptide 3412 should have a charge of +3, peptide 3413 +2, peptide 314 +3 and so on.

Thanks for the warning, data in Table 2 were corrected following the instructions.

Table 4 should also be corrected, because the authors have never tested the activity of the GWFSLVRKVVGGVGSL peptide and the GWFSLVRKVVGGVGSL-NH2 peptide. So either this Table should be deleted or the values given there corrected (molar mass, chemical formula, predicted isoelectric point, GRAVY index and stability index). The authors do not seem to understand the difference between a peptide with a free carboxyl group and a peptide with a C-terminal carboxamide group.

Thanks for the warning, data in Table 4 was corrected following the instructions. Stability index was deleted, because the predictor that was used did not calculate it with the amidation on C-terminal.
